# Screening for Major Depressive Disorder Using a Wearable Ultra-Short-Term HRV Monitor and Signal Quality Indices

**DOI:** 10.3390/s23083867

**Published:** 2023-04-10

**Authors:** Shohei Sato, Takuma Hiratsuka, Kenya Hasegawa, Keisuke Watanabe, Yusuke Obara, Nobutoshi Kariya, Toshikazu Shinba, Takemi Matsui

**Affiliations:** 1Department of Electrical Engineering and Computer Science, Faculty of Systems Design, Tokyo Metropolitan University, Tokyo 191-0065, Japan; 2Maynds Tower Mental Clinic, Tokyo 151-0053, Japan; 3Department of Psychiatry, Shizuoka Saiseikai General Hospital, Shizuoka 422-8527, Japan; 4Research Division, Saiseikai Research Institute of Health Care and Welfare, Tokyo 108-0073, Japan; 5Department of Electrical Engineering and Computer Science, Graduate School of System Design, Tokyo Metropolitan University, Tokyo 191-0065, Japan

**Keywords:** major depressive disorder, ultra-short-term heart rate variability, autonomic nervous response, photoplethysmography, signal quality index, machine learning

## Abstract

To encourage potential major depressive disorder (MDD) patients to attend diagnostic sessions, we developed a novel MDD screening system based on sleep-induced autonomic nervous responses. The proposed method only requires a wristwatch device to be worn for 24 h. We evaluated heart rate variability (HRV) via wrist photoplethysmography (PPG). However, previous studies have indicated that HRV measurements obtained using wearable devices are susceptible to motion artifacts. We propose a novel method to improve screening accuracy by removing unreliable HRV data (identified on the basis of signal quality indices (SQIs) obtained by PPG sensors). The proposed algorithm enables real-time calculation of signal quality indices in the frequency domain (SQI-FD). A clinical study conducted at Maynds Tower Mental Clinic enrolled 40 MDD patients (mean age, 37.5 ± 8.8 years) diagnosed on the basis of the *Diagnostic and Statistical Manual of Mental Disorders, Fifth Edition*, and 29 healthy volunteers (mean age, 31.9 ± 13.0 years). Acceleration data were used to identify sleep states, and a linear classification model was trained and tested using HRV and pulse rate data. Ten-fold cross-validation showed a sensitivity of 87.3% (80.3% without SQI-FD data) and specificity of 84.0% (73.3% without SQI-FD data). Thus, SQI-FD drastically improved sensitivity and specificity.

## 1. Introduction

Major depressive disorder (MDD) is a severe mental illness that affects many people, and can result in self-injurious behavior and suicide. According to the World Health Organization, approximately 280 million people worldwide experienced depression in 2021 [1]. Moreover, according to one study, MDD is the second most serious health issue worldwide with respect to disability-adjusted life years [2]. Screening for depression in adults, including elderly and pregnant populations, is recommended in the United States because of the effectiveness of medications and cognitive behavioral therapy for prevention and early intervention [3,4]. 

Clinicians typically diagnose MDD according to the guidelines provided by the *Diagnostic and Statistical Manual of Mental Disorders, Fifth Edition* (*DSM-5*) [5]. Currently, the benefit of brief screening using questionnaires is considered questionable [6]. Although quantitative diagnostic methods for MDD are needed, there is a lack of evidence supporting the diagnosis of MDD on the basis of physiological biomarkers [7,8]. Recently, studies of digital biomarkers have been performed using data obtained from smartphones and wearables [9,10,11,12]. Given the widespread use of these commercially available devices, this novel screening approach is expected to facilitate early intervention and prevention for many potential MDD patients. Wearables noninvasively provide continuous 24 h multimodal data, such as electrocardiography (ECG), actigraphy, pulse rate, sleep duration, perspiration, and peripheral skin temperature data [13,14,15,16,17]. Automatic methods for screening and assessing the severity of MDD have been investigated, and have also been applied to detect stressors under free-living conditions [18,19,20].

Heart rate variability (HRV) refers to the temporal variability between consecutive heartbeat intervals [21], and reflects the activity of the sympathetic and parasympathetic nervous systems; these systems regulate respiratory sinus arrhythmias and the cardiovascular system. HRV is a useful indicator of mental stress, and resting HRV has been shown to be significantly decreased in patients with MDD [22,23]. Activity in the high-frequency (HF) band is lowered by stressful stimuli and anxiety [24,25,26,27]. Decreased HRV in MDD patients increases the risk of cardiovascular disease, and HRV may serve as a crucial indicator of the treatment response [22,28,29]. Shinba et al. proposed a method to screen MDD patients by focusing on transient changes in HRV before and after mental stress induced by a random number generation task; since HRV exhibits considerable interindividual variation, more reliable screening results can be obtained by analyzing the transient response to applied stress rather than HRV at rest [25,30,31]. Using the Ewing test to assess the autonomic transient response, Kuang et al. obtained similar results [32]. The golden standard for HRV measurement is ECG. However, HRV can also be measured with sufficient accuracy using photoplethysmography (PPG) sensors under static conditions [33,34] and at night [35], although this is not possible during physical activity or in the presence of certain mental stressors [36]. Dagdanpurev et al. and Unursaikhan et al. showed that an autonomic transient response-based MDD screening method provided valid data even when using PPG signals extracted from finger pulse waves [37] and webcams [38], respectively. These studies suggest that MDD screening can be performed using wearable devices equipped with PPG sensors.

PPG sensors are noninvasive and capture pulse-dependent increases or decreases in the light absorbance of hemoglobin via transmitted or reflected light-emitting diode light irradiating the skin [39,40]. Pulse oximeters are commonly used in hospitals to obtain PPG data. Accurate peak positions and pulse waveforms are needed to extract HRV data, as well as data for other cardiovascular parameters such as blood pressure, from PPG waveforms [40,41]. Because motion artifacts affect the data obtained in uncontrolled environments (e.g., under continuous 24 h monitoring conditions), signal quality assessment is essential. Previous studies developed methods to detect motion artifacts in PPG signals, as well as signal quality degradation due to ambient light, using signal quality indices (SQIs) such as kurtosis and skewness [42]. Such SQIs aid in the detection of motion artifacts using a simple algorithm, but cannot reveal whether the shape of the pulse waveform has been sufficiently conserved. Another study proposed a method to remove unreliable data using wavelet transforms; the method improved the accuracy of peak-to-peak interval (PPI) and time-domain HRV indices [43].

Pedrelli et al. assessed the severity of MDD using the E4 wristband (Empatica, Cambridge, MA, USA), a smartphone sensor, and usage data collected over 8 weeks [9]. The wristband data included electrodermal activity, peripheral skin temperature, and pulse rate, but not HRV. According to a regression model, the mean prediction error for the 17-item Hamilton Depression Rating Scale ranged from 3.88 ± 0.18 to 4.74 ± 1.24. Tazawa et al. assessed the severity of MDD by analyzing hourly step count, energy expenditure, body movement, sleep duration, pulse rate, skin temperature, and ultraviolet exposure data obtained by the Silmee W20 wristband (Toshiba, Tokyo, Japan) [44]. The accuracy rate of their classification model for symptomatic MDD patients was 76%.

In the studies mentioned above that performed MDD screening using wearable devices in free-living conditions, HRV was either not used at all or did not play a major role. In another study, the screening accuracy was 71% despite the use of HRV [45]. There are two possible reasons for this. First, under free-living conditions, although resting periods can be identified using an accelerometer, detecting stressful events, i.e., events that can trigger autonomic response, is challenging. Second, PPG sensors are sensitive to body motion. If MDD screening is performed on the basis of HRV, classification will be compromised unless a large amount of training PPG data of sufficient accuracy are available. To address these issues, we screened for MDD on the basis of reliable HRV features obtained before, during, and after sleep. Here, a sleep relaxation intervention was used instead of a task-based intervention evoking autonomic nerve activity, such as random number generation tasks, which are conventionally employed. In addition, the PPG signal was filtered using signal quality indices in the frequency domain (SQI-FD) that represent the shape of frequency spectra. Although using SQIs reduces the amount of data available for 24 h HRV analysis, our framework compensates for this by using ultra-short-term heart rate variability (USTHRV), i.e., HRV calculated from PPG signals shorter than 5 min (overlapping sliding window of 1 min). Previous studies on USTHRV have demonstrated that frequency domain HRV indices require longer ECG/PPG records than time-domain HRV indices to ensure accurate results. In this study, USTHRV with 3 min intervals was used [46,47]. Under this condition, the frequency-domain HRV indices would have a correlation coefficient of ≥0.9, with short-term HRV having 5 min intervals. In 24 h HRV recording data, both short-term and long-term HRV analyses are applicable [48]. Short-term HRV analysis, which analyzes 5 min subsegments, is more prevalent but susceptible to noise; furthermore, the physiological significance of these indices may be ambiguous when measured in free-living conditions [49]. Long-term HRV analysis can mitigate the effects of noise and changes in HRV over time while enabling the acquisition of more stable results for the very-low-frequency component. This study employed USTHRV as a surrogate for short-term HRV to capture dynamic changes in HRV indices over time. The proposed method, applicable to short-duration data, is advantageous because standard wearable devices, unlike 24 h Holter ECGs, have limited battery capacity and necessitate intermittent measurements.

We trained logistic regression models using HRV indices calculated with the proposed framework. By applying 10-fold cross-validation, we evaluated classification performance using the data of 40 unipolar MDD patients and 29 healthy adults. Then, 24 h measurements obtained using a wristwatch device identified MDD patients with a sensitivity of 87.3% and specificity of 84.0%. Furthermore, performance was significantly superior when SQI-FD data were analyzed. For MDD screening, signal quality assessment is recommended when measuring HRV under free-living conditions using wristband devices.

## 2. Proposed Framework

The 24 h HRV analysis framework is shown in Figure 1; it was implemented using Python libraries (NumPy [50] and SciPy [51]). Using the 24 h continuous PPG and 3-axis acceleration data obtained by the E4 wristband as input, sleep duration, pulse rate, and HRV data were derived at 1 min intervals (1 epoch = 1 min). Sleep and wake sections were discriminated according to activity amplitude, which was derived from the acceleration data using a traditional algorithm. The MUSIC algorithm was employed for pulse rate detection. An SQI for pulse rate detection (SQI_pr_) was used to skip calculations at epochs where accurate pulse rate estimation was deemed unlikely to be accurate. For epochs where pulse rates were successfully obtained, HRV was calculated. Again, PPG data with low signal quality were removed in advance on the basis of the signal quality index for heart rate variability (SQI_hrv_). HRV indices in the time and frequency domains were compared between healthy adults and MDD patients, and logistic regression was applied for binary classification.

### 2.1. Sleep/Wake Estimation

Accelerations sampled at 32 Hz were bandpass-filtered using a Butterworth filter (0.25–3.0 Hz), and activity amplitudes were calculated for each epoch using the Scripps Clinic algorithm [52], which is a traditional algorithm for distinguishing sleep and waking states using wrist actigraphy. In an earlier study, the rescored Scripps Clinic algorithm had the highest F1 score and second-highest accuracy among all traditional algorithms used for sleep/wake classification [53].

Traditional algorithms tend to overestimate sleep duration [53]. We included bed rest before and after sleep as part of the intervention period because our MDD screening method considers the relaxed state to reflect sympathetic dominance. Rescoring should conservatively correct for “false” sleep epochs such that detection sensitivity is not significantly reduced. In cases where multiple segments of sleep are detected within 24 h, HRV analysis is performed with a focus on the longest sleep segment. To achieve this, we implemented the following rescoring steps, which differ from those proposed by Webster et al. and Cole et al. [54]:Change the status from “sleep” to “awake” of <120 consecutive minutes between two epochs with activity amplitudes above the 75th percentile value for all epochs.Among connected “sleep” fragments, which can include up to 100 min of mid-waking, only the longest segment is retained; the others are changed to “awake.”Find the beginning and end of the fragment of consecutive “sleep,” which is longest when awakenings <4 min are ignored. Any fragments of “sleep” <60 min at both ends of that are corrected to “awake.”

### 2.2. Pulse Rate Detection with Thresholding Using SQIpr

As well as HRV, pulse rate reflects the status of the sympathetic and parasympathetic nervous systems. Because pulse rate is also a proxy for the mean PPI and can enhance the accuracy of HRV calculations, the heart rate is calculated before HRV analysis. The proposed algorithm is based on the subspace MUSIC algorithm [55], which is used to detect the pulse rate. The subspace-based method is more robust to noise and has a higher frequency resolution than fast Fourier transform. However, it requires more computation time than conventional methods because matrix calculations are performed repeatedly. Therefore, SQI_pr_ was used to filter out PPG data, for which accurate pulse rate detection is difficult when using the MUSIC method. It should be noted that this index is intended to improve calculation efficiency and does not enhance the accuracy of pulse rate detection or HRV analysis.

Our proposed SQI for pulse rate detection, SQI_pr_, was calculated in the frequency domain. We used Welch’s method [56] to estimate power spectral density (PSD). First, raw PPG signals sampled at 64 Hz by the E4 sensor were bandpass-filtered at 0.5–5 Hz. The initial pulse rate estimate was considered as the frequency bin with the maximal peak in PSD in the 40–160 bpm range. The standard deviation for all frequency bins >−5 dB relative to the maximum was calculated. The target range for pulse rate detection was 40–160 bpm and no more than twice the initial pulse rate estimate. Using this standard deviation, SQI_pr_ was defined (Algorithm 1).

For each epoch, SQI_pr_ was calculated using a 3 min sliding window (1 min overlap) before and after each epoch. The Blackman–Harris window was applied for the PSD calculation. Pseudo-spectra were calculated by the MUSIC algorithm for all epochs in which the SQI_pr_ was above the threshold value. In the pseudo-spectra, the maximum peak (in the range of 40–120 bpm in the sleep state and 40–160 bpm in the waking/resting states) was used for the final pulse rate estimate. The threshold value for SQI_pr_ was set at 0.5, and an activity amplitude of ≤300 was defined as the resting state. Data from non-resting epochs were not used for HRV analysis because prior studies did not recommend that USTHRV analysis be performed under non-resting conditions [57]. In epochs with missing pulse rate data, median-filtered interpolation of the pulse rate was performed using a 7 min data window.
**Algorithm 1:** Calculation of SQI_pr_**Input**: Spectral density *dn* and Frequency bin *bin* of PSDTarget range *f*_min_ and *f*_max_**Output**: *SQI*_pr_1:  *f*_init_ = *bin*[argmax(*dn*)]2:  *threshold* = *dn*[*f*_init_]/3.16    //−5 dB3:  *idx_array* = where *dn* > *threshold* in range [*f*_min_, max (*f*_max_, 2.0 * *f*_init_)]4:  **if** length of *idx_array* > 1 **then**5:     *bin_std* = standard deviation of *bin*[i*dx_array*]6:     **return**  1.0–3.0 * *bin_std*/(max (*f*_max_, 2.0 * *f*_init_ ) − *f*_min_)7:  **else**8:     **return**  1.0

### 2.3. USTHRV Analysis with Thresholding Using SQI_hrv_

As in the previous subsection, 3 min sliding windows were applied to compute another SQI and HRV for each epoch. SQI_hrv_ is the ratio between the harmonic components up to the third order and inharmonic components in the frequency spectrum of the PPG signal. The PPG waveform depends on multiple factors in the circulatory system and skin. Because it is unusual for the waveform of a person without arrhythmia to change significantly within a few minutes at rest, we assumed that the signal was periodic, including the harmonics. For PPG data with high signal quality, i.e., where the pulse waveform retains information such as the peak positions, energy is concentrated in the heartbeat component and its harmonics, as shown in Figure 2 (left). However, for data whose quality has been degraded by motion artifacts and other types of noise, the energy of the non-integer components increases to a greater extent, as also shown in Figure 2 (right). SQI_hrv_ is calculated using the algorithm shown in Algorithm 2. The same PSD data in RAM used for calculating SQI_pr_ can be used again for each epoch. The fundamental frequency is derived using the MUSIC algorithm described in the previous subsection. Employing a robust frequency estimator prevents the calculation of inaccurate values for the signal quality, even for noisy PPG data.
**Algorithm 2:** Calculation of *SQI*_hrv_**Input:** Spectral density *dn* and Frequency bin *bin* of PSDFundamental frequency of heartbeats *f*_MUSIC_Margin in frequency bin *f*_margin_**Output**: *SQI*_hrv_1:  *idx_h1* = *bin* in range [*f*_MUSIC_ − *f*_margin_, *f*_MUSIC_ + *f*_margin_]2:  *idx_h2* = *bin* in range [2.0**f*_MUSIC_ − *f*_margin_, 2.0**f*_MUSIC_ + *f*_margin_]3:  *idx_h3* = *bin* in range [3.0**f*_MUSIC_ − *f*_margin_, 3.0**f*_MUSIC_ + *f*_margin_]4:  *idx_i1* = *bin* in range [0.5**f*_MUSIC_ + *f*_margin_, *f*_MUSIC_ − *f*_margin_]5:  *idx_i2* = *bin* in range [*f*_MUSIC_ + *f*_margin_, 2.0**f*_MUSIC_ − *f*_margin_]6:  *idx_i3* = *bin* in range [2.0**f*_MUSIC_ + *f*_margin_, 3.0**f*_MUSIC_ − *f*_margin_]7:  *A*_harmonic_ = sum of *dn*[*idx_h1*] + sum of *dn*[*idx_h2*] + sum of *dn*[*idx_h3*]8:  *A*_non-harmonic_ = sum of *dn*[*idx_i1*] + sum of *dn*[*idx_i2*] + sum of *dn*[*idx_i3*]9:  **if** *A*_non-harmonic_ > 0 **then**10:        **return**  log(1.0 + *A*_harmonic_ /*A*_non-harmonic_)11: **else**12:        **return**  0.0

All epochs for which SQI_hrv_ was above the threshold value were preprocessed using known pulse rates (*f*_MUSIC_; Algorithm 2). The threshold value of SQI_hrv_ was empirically determined as 1.0 on the basis of some example data. The raw PPG data were 4× upsampled [58,59] and the first-order difference operation was applied, followed by a bandpass filter (range: 0.75*f*_MUSIC_ to 6.5*f*_MUSIC_). Peak detection was performed using the *find_peaks* function in the SciPy library. To remove the second peaks in the double-peak PPG waveform, we first removed all peaks in which one previous valley was >0. The PPI used for HRV analysis was calculated using the obtained peaks. Simple validation was performed by comparing the number of measured peaks with the number calculated from the data length and pulse rate, and the mean PPI measured with the reciprocal of the pulse rate; if either value was 30% higher or lower than the other, HRV analysis was not performed for that epoch. We did not interpolate for epochs with no HRV data.

The HRV indices used for MDD screening in this study are shown in Table 1. Although HRV indices characteristic of MDD have been identified at rest and in response to mental stress, few studies have validated their use in relaxation tasks. Thus, we selected several indices commonly used in previous methods. The frequency-domain indices were calculated using Welch’s method.

## 3. Experimental Procedure

We evaluated the proposed algorithm using the data of 40 MDD patients (18 males and 22 females) diagnosed by clinicians using the *DSM-5* at Maynds Tower Mental Clinic (Tokyo, Japan), as well as the data of 29 healthy adult volunteers (15 males and 14 females) recruited at Tokyo Metropolitan University (Tokyo, Japan). All participants were legal adults and provided written informed consent. They were instructed not to consume alcohol or caffeine for 24 h before the study, and to refrain from smoking on the day of the study. We assessed symptom severity in both groups using the Zung Self-Rated Depression Scale (SDS). The MDD patients had not been diagnosed with a physical illness and were participating in a return-to-work program at the time of the study. This study was approved by the Ethics Committee of Tokyo Metropolitan University (approval number: 411) and was conducted from October 2019 to September 2021. The participants’ demographic and clinical characteristics are summarized in Table 2.

The E4 wristband can continuously record acceleration, PPG, electrodermal activity, and skin surface temperature data for >32 h. Our participants wore the E4 device on their non-dominant wrist; data were recorded for 24 h under free-living conditions. They were asked to record activities such as sleeping, eating, and bathing using a log sheet.

The proposed algorithm processed the 24 h sensor data, and the HRV analysis focused on sleep/relaxation periods, as shown in Figure 3. We established the phases shown in Figure 3 as the unit of HRV analysis. Each phase was set to a duration of 90 min, based on the standard sleep cycle for adults, except P_4_, which has a variable length to accommodate individual differences in total sleep time (TST). P_1_ corresponds to the awake period immediately prior to sleep onset, while P_2_ represents the initial period following sleep onset. A similar procedure was employed to define the five phases up to P_5_, which is the first period after awakening. This method requires at least 270 min of sleep time data per participant; one MDD patient was excluded from the analysis because their TST was below this threshold. Participants with no data suitable for analysis after SQI-FD thresholding (because of persistently low signal quality; one healthy adult and one MDD patient) were also excluded from the analysis.

Logistic regression was performed to discriminate between healthy adults and MDD patients on the basis of 20 features and the mean values of the HRV indices in each phase (two-class classification). For pipeline processing, we implemented 10-fold cross-validation, standardized scaling, and logistic regression using the Scikit-Learn library [60]. The training/validation/test split occasionally provided inconsistent results when applied to holdout samples due to inadequate sample sizes. Moreover, when employing leave-one-out cross-validation, the classification accuracy of the training model tended to be overestimated. Because some HRV indices show multicollinearity, L2 penalty terms were added separately to the logistic regression model.

## 4. Results

### 4.1. Sleep Time Estimation

The TST estimated by the proposed algorithm is plotted against the self-reported sleep duration (as recorded in the log sheet) in Figure 4. The self-reported sleep duration is a subjective value derived from manual recordings made every 15 min; the mean absolute percentage error was 13.6% in the healthy group and 15.3% in the MDD group. The TST was underestimated by >60 min in only one MDD patient.

### 4.2. SQI Thresholding and HRV Indices

The percentages of data that remained after thresholding using the SQI-FD are shown in Table 3. More than 70% of the awake-period data were filtered out by SQI-FD, i.e., were not included in the HRV analysis, whereas most of the sleep/rest period data were free from motion artifacts. The computer processing unit (CPU) time was measured using a MacBook Air with 16 GB of RAM. Using our algorithm, the mean CPU time was reduced by 45.9% because data with low signal quality were skipped.

Table 4 shows the mean values of the HRV indices in each phase obtained after applying the proposed algorithm. For phases P_2_–P_4_, HRV data from epochs classified by the Scripps Clinic algorithm as awake after sleep onset were excluded from the mean calculations. In four participants with MDD (participants 60, 61, 71, and 73), the number of epochs available for HRV analysis was zero in at least one phase after thresholding by SQI-FD. Data from these participants were not included in the analyses that yielded the following results. The values for most HRV indices tended to decrease after being filtered by SQI_hrv_. The Wilcoxon signed-rank test showed significant differences in all indices, except the low-frequency (LF)/HF ratio, with versus without thresholding; HRV indices that were not significantly different are underlined in the table. The Mann–Whitney U test revealed significant differences in at least one phase between the healthy adults and MDD patients for all HRV indices.

The mean root mean square of successive differences between normal heartbeats (RMSSD), standard deviation of N-N intervals (SDNN), LF, and total power (TP) values were lower in the MDD patients than healthy adults, and a U-shaped response was seen (i.e., the values decreased during sleep and returned to normal upon awakening). The LF/HF ratio was significantly higher in the MDD than healthy adult group in P_3_ (*p* < 0.001).

### 4.3. MDD Screening

For binary classification, a logistic regression model that included HRV indices and pulse rate was used. The HF, LF, and LF/HF values were analyzed, but the RMSSD, SDNN, and TP values were omitted because they were strongly correlated with the sum of HF and LF [61] and the variance inflation factor (VIF) was high (>100) [62]. LF and HF, which tend to be log-normally distributed, were log-transformed before the analysis. All of the VIFs in the feature set were <10. Grid searches were performed for each condition (with and without SQI-FD) to optimize the L2 regularization strength *C*.

The receiver operating characteristic curve obtained through 10-fold cross-validation is shown in Figure 5. The area under the curve (AUC) values were 0.93 (95% confidence interval (CI): 0.86–0.99) and 0.81 (95% CI: 0.71–0.92) with and without the application of SQI-FD thresholding, respectively.

Figure 6 is a scatterplot of the SDS scores against the logit scores for the trained model, with SQI-FD thresholding applied. There was a positive correlation between the two scores (*r* = 0.55), which was higher than in our previous study [30]. The optimal SDS cutoff score for MDD was 40 (sensitivity = 0.784, specificity = 0.793). This threshold shows good agreement with previous studies [63,64].

Table 5 shows the average classification performance after 100 iterations of 10-fold cross-validation (random sampling). The classification accuracy was highest when SQI-FD thresholding was applied to both the awake and sleep periods, followed by when it was applied only to the awake period.

## 5. Discussion

Wearable devices enhance privacy and confidentiality because users collect their own personal data. It is hoped that noninvasive, unrestrained, objective MDD screening can be achieved using wearable devices, which may also promote early medical consultations and self-management. In particular, the data shown in Figure 6 related to MDD risk can be easily understood by patients and will benefit those seeking to return to work after consultation and treatment [65,66].

Obtaining indices of HRV can aid screening for MDD because of the tendency toward dysautonomia of sufferers [67]. Indeed, HRV can be measured to track autonomic responses to mental stress over a short period [31,32]. Moreover, daily tracking allowed an MDD screening accuracy of >70% to be achieved [45,68]. However, two issues make it challenging to perform HRV measurements under free-living conditions: the deterioration of sensor signal quality over time because of body motion and the difficulty of determining the analysis period for transient autonomic responses.

In the HRV indices shown in Table 4, a common trend can be observed: MDD patients exhibit higher LF/HF and lower other HRV indices compared with healthy adults. These findings align with those of previous studies [22,69]. All HRV indices, except LF/HF in MDD patients, demonstrate a V-shaped temporal transition, higher in the awake period and lower during sleep. SDNN, LF, and LF/HF are also consistent with a previous study [67], as these indices are lower during sleep than in awake periods, except for rapid eye movement (REM) sleep. It is important to note, however, that this study did not measure sleep stages, and all valid data were averaged for each phase; thus, the results are only consistent on average.

Research has indicated that HF increases during the night in healthy young adults [68]. In this study, although the percent power in the HF band increased similarly to the literature, HF tended to decrease during sleep. We attribute this to the fact that the effect of body movement is more dominant in awake data than in sleep data; the diminishing of this trend after thresholding with the SQI-FD supports this assertion. Nonetheless, the HF in sleep after noise removal did not increase, and the trend remained flat, suggesting that the filtered data was not entirely free of motion artifacts. The HF HRV components exhibited the most significant difference between healthy adults and MDD patients during relaxation, particularly after more than 3 h from sleep onset (P_3_ and P_4_). Our findings suggest that the parasympathetic activity associated with sleep/relaxation may be suppressed in MDD patients.

According to the data in Table 5, the proposed method distinguished MDD patients from healthy controls with a sensitivity of 87.3% and specificity of 84.0%, and the classification performance was superior to that achieved using Zung’s SDS score with optimal thresholding. Table 4 shows that the HRV indices were lower with versus without SQI-FD thresholding. Measurements affected by motion artifacts usually overestimate HRV indices [36], suggesting that our SQI-FD thresholding process may have removed HRV data affected by body motion. These results indicate the importance of removing unreliable data when screening for MDD; although we had expected that to be the case for waking-state data, Table 5 shows that it also applies to sleep-state data.

To assess autonomic transient responses under free-living conditions, i.e., without the imposition of mental stress, HRV indices measured by a wearable device must be analyzed in the context of the user’s typical behavior. In this study, individual differences in pre- and post-sleep activity may have affected the results, and misclassification of sleep and waking states might have occurred; however, the impact of these factors on screening performance did not appear to be critical. The analysis scheme shown in Figure 3 can be used for MDD screening with acceptable accuracy, even when using a wrist actigraphy system that automatically detects sleep.

There were some limitations to this study. First, when setting the threshold for SQI-FD, most of the waking period data were omitted, and HRV analysis could not be performed in four subjects who had no data. To reliably screen for MDD over a single 24 h period, a more flexible thresholding process is needed to ensure that sufficient data with acceptable signal quality are collected [70]. To prevent unnecessary discarding of data containing noise, preprocessing should be performed to correct sections for PPG data with motion artifacts, and HRV analysis should be performed with variable epoch lengths. As a second limitation, our MDD patients were all taking antidepressants; previous studies have shown that antidepressants may decrease HRV [24,71]. Although the proposed method focuses on HRV changes associated with sleep relaxation, and may be less susceptible to small increases or decreases in HRV, future studies should enroll patients who are not taking antidepressants. Third, the dataset used in this study was small (69 participants) compared with the datasets typically analyzed in medical informatics studies; moreover, the reliability of evaluations using leave-one-out cross-validation and holdout methods has been questioned, and it is important to evaluate data that have not been used in the training and hyperparameter tuning processes. Finally, our measurements were obtained under free-living conditions, such that various confounders may have influenced the experimental results. In the future, we plan to analyze a larger amount of clinical data obtained with the wearable devices.

## Figures and Tables

**Figure 1 sensors-23-03867-f001:**
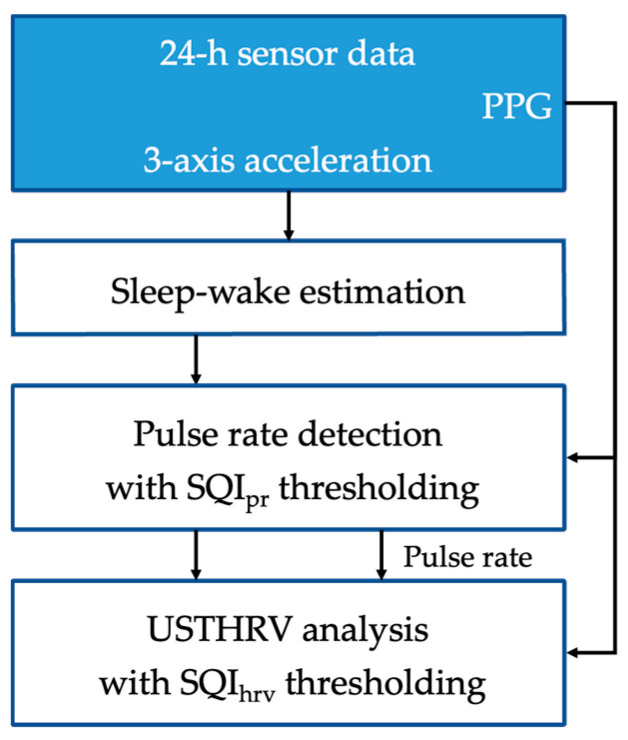
The 24 h heart rate variability analysis framework.

**Figure 2 sensors-23-03867-f002:**
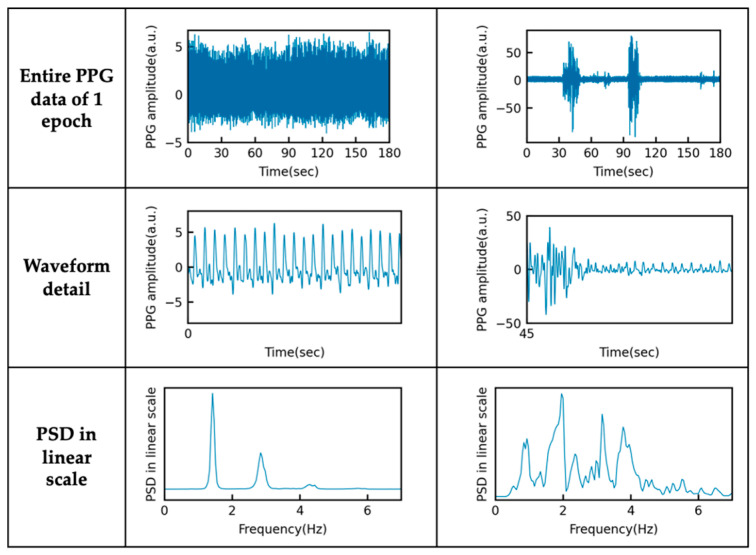
Photoplethysmography (PPG) data and the associated power spectral density (PSD) shape during sleep in a single major depressive disorder patient (ID = 45). Left: 572nd epoch, signal quality index for heart rate variability (SQI_hrv_) = 2.6; right: 552nd epoch, SQI_hrv_ = 0.3.

**Figure 3 sensors-23-03867-f003:**
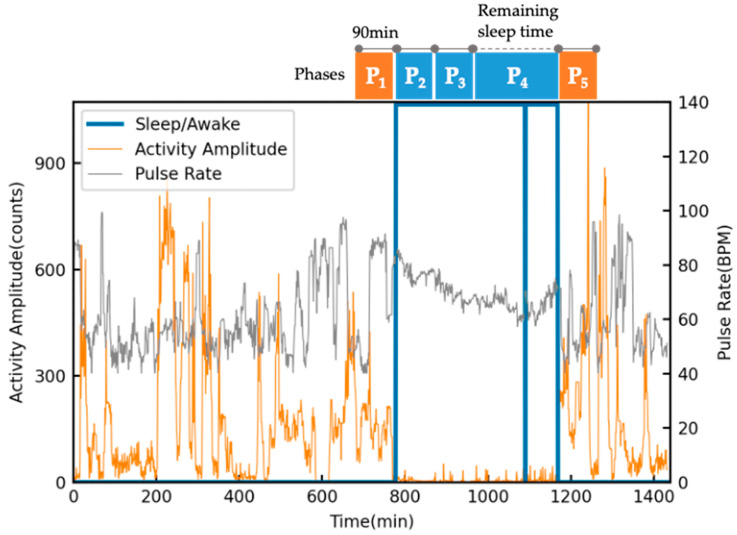
The five phases for heart rate variability analysis in the proposed method.

**Figure 4 sensors-23-03867-f004:**
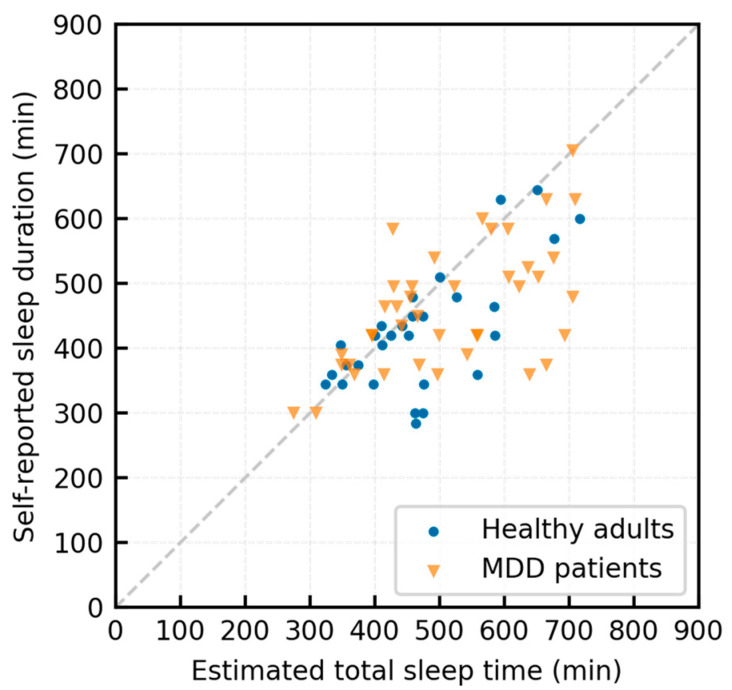
Scatterplot of the estimated total sleep time and self-reported sleep duration (as recorded in log sheets).

**Figure 5 sensors-23-03867-f005:**
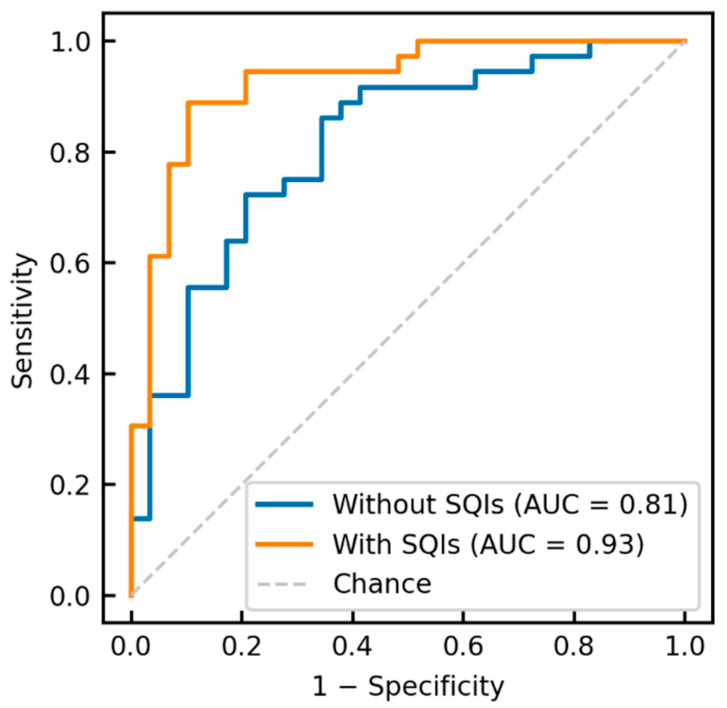
Receiver operating characteristic curves derived from logistic regression with 10-fold cross-validation.

**Figure 6 sensors-23-03867-f006:**
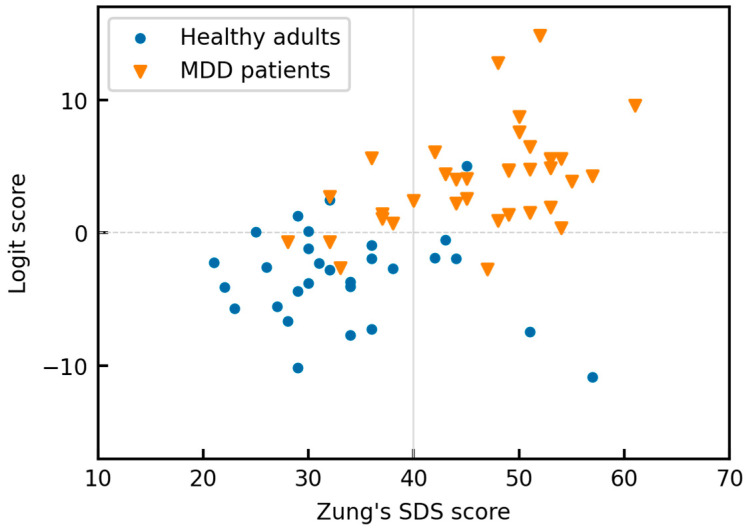
Scatterplot of the Zung Self-Rated Depression Scale (SDS) scores against the logit scores obtained using the proposed algorithm.

**Table 1 sensors-23-03867-t001:** Heart rate variability indices used in this study.

Method	Index	Unit	Description
Time domain	RMSSD	ms	Root mean squared successive differences of PPI
	SDNN	ms	Standard deviation of PPI
Frequency domain	LF	ms^2^	Absolute power of the low-frequency band (0.04–0.15 Hz)
	HF	ms^2^	Absolute power of the high-frequency band (0.15–0.40 Hz)
	LF/HF		Ratio between LF and HF
	TP	ms^2^	Total power of all frequency bands

**Table 2 sensors-23-03867-t002:** Demographic and clinical characteristics of the study groups.

	Total	Healthy Adults	MDD Patients	*p*-Value
*n*	69	29	40	
Male (%)		15 (51.7)	18 (45.0)	N.S. (χ^2^)
Female (%)		14 (48.3)	22 (55.0)	
Age in years, mean (SD)	35.6 (11.3)	31.9 (13.0)	37.5 (8.8)	N.S.
Self-reported sleep duration (min)	448.0 (95.8)	426.7 (94.3)	463.5 (95.0)	N.S.
SDS scores, mean (SD)	40.6 (10.2)	34.0 (8.6)	45.7 (8.2)	<0.001

**Table 3 sensors-23-03867-t003:** Percentages of data remaining and computer processing unit time after thresholding using signal quality indices in the frequency domain.

	**Total (%)**	**Healthy Adults (%)**	**MDD Patients (%)**
Awake period			
SQI_pr_	41.1	46.0	36.9
SQI_pr_ and SQI_hrv_	22.7	26.1	19.8
Sleep period			
SQI_pr_	88.0	87.5	88.4
SQI_pr_ and SQI_hrv_	83.9	84.5	83.4
	**Without SQI_pr_**	**With SQI_pr_**	**Reduction**
Mean CPU time (s)	139.96	79.47	45.9%
SD	5.4	19.2	13.3%

**Table 4 sensors-23-03867-t004:** Differences in HRV indices with and without thresholding by real-time calculation of signal quality indices in the frequency domain (SQI-FD).

HRV Index	Phase	Healthy Adults		MDD Patients	
		Without SQI-FD	With SQI-FD	Without SQI-FD	With SQI-FD
RMSSD (ms)	P_1_	100.6	84.5	91.2	69.0 **
	P_2_	62.1	60.0	52.1 *	46.4 **
	P_3_	65.5	64.1	46.0 **	40.4 ***
	P_4_	72.3	70.0	50.9 ***	48.0 ***
	P_5_	99.4	81.5	92.7	70.8 ***
SDNN (ms)	P_1_	91.8	78.2	79.6 **	62.6 ***
	P_2_	64.0	60.6	53.3 *	46.4 **
	P_3_	64.4	61.3	54.1	47.8 **
	P_4_	72.9	69.5	57.9 **	53.8 ***
	P_5_	94.0	80.6	83.1 ***	65.4 ***
LF (ms^2^)	P_1_	1727	1369	1006 ***	689 ***
	P_2_	850	785	511 *	417 **
	P_3_	718	660	540	444 *
	P_4_	981	911	548 **	492 ***
	P_5_	1594	1276	1119 ***	726 ***
HF (ms^2^)	P_1_	1833	1326	1229 **	748 ***
	P_2_	1045	1014	625 **	511 **
	P_3_	1279	1257	497 ***	405 ***
	P_4_	1426	1386	586 ***	550 ***
	P_5_	1653	1208	1429 **	876 ***
LF/HF	P_1_	1.22	1.41	1.20	1.41
	P_2_	1.25	1.21	1.50	1.52
	P_3_	0.98	0.93	1.72 **	1.72 ***
	P_4_	1.13	1.11	1.44	1.41
	P_5_	1.19	1.40	1.12	1.22
TP (ms^2^)	P_1_	4175	3139	2977 **	1834 ***
	P_2_	2275	2130	1519 *	1217 **
	P_3_	2391	2270	1354 **	1061 **
	P_4_	2863	2710	1531 ***	1384 ***
	P_5_	3862	2804	3162 **	1891 ***

* *p* < 0.05, ** *p* < 0.01, *** *p* < 0.001

**Table 5 sensors-23-03867-t005:** Major depressive disorder screening performance with and without thresholding using signal quality indices in the frequency domain (SQI-FD).

	Accuracy	Sensitivity	Precision	NPV	F1 Score	MCC
Without SQI-FD	0.772	0.803	0.733	0.750	0.796	0.537
With SQI-FD	0.859	0.873	0.840	0.842	0.872	0.714
Awake period	0.816	0.834	0.794	0.794	0.834	0.628
Sleep period	0.798	0.828	0.760	0.781	0.819	0.590

## Data Availability

The data are not publicly available due to privacy and ethical restrictions.

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
