# Peer review of "Screening for Major Depressive Disorder Using a Wearable Ultra-Short-Term HRV Monitor and Signal Quality Indices"

_sensors, 2023, doi:10.3390/s23083867_

Round 1
Reviewer 1 Report
The authors evaluated major depressive disorder using a wearable ultrashort-term HRV and signal quality indices. Their main idea is the noise removal based on signal quality indices to obtain reliable PPG signals to calculate major depressive disorder in daily life. My main concerns and questions are as follows:
1. They proposed ultrashort-term HRV instead of long-term HRV to evaluate major depressive disorder. What is the advantages of the ultrashort-term HRV and how can demonstrate it? It is necessary to describe background research and validate it by comparing ultrashort-term HRV with long-term HRV in your experimental data.
2. Are the results of HRV on MDD patients valid compared to other research on MDD? In addition to classification accuracy by machine learning, physiological interpretation is required.
Author Response
We sincerely appreciate your thoughtful and constructive feedback on our manuscript. In response to the comments and advice provided, we have carefully revised the manuscript and offer the following detailed responses to address your queries.
Reviewer #1:
Point 1:
They proposed ultrashort-term HRV instead of long-term HRV to evaluate major depressive disorder. What is the advantages of the ultrashort-term HRV and how can demonstrate it? It is necessary to describe background research and validate it by comparing ultrashort-term HRV with long-term HRV in your experimental data.
Response:
We sincerely appreciate the constructive feedback on our manuscript. Based on the feedback, we have made the following revisions to improve the quality and clarity of our work:
In the current manuscript, we have added 130-140 lines on page 3 to describe our approach and comparison between long-term HRV analysis. We have incorporated additional references to the relevant literature to strengthen our arguments.
Our work specifically addressed the limitations of short-term and ultrashort-term heart rate variability (HRV) indices for 24-h recording data, which are known to be sensitive to signal quality degradation. Long-term HRV analysis can overcome this shortcoming. However, some temporal resolution is required to analyze the dynamics of the autonomic nervous system's response to mental load, as we aimed to do in this study. However, previous studies have shown that long-term trends can disappear when short-term analysis results are averaged, so comparing and combining both is important.
This study contributes to the field by alleviating the limitations of short-term and ultrashort-term HRV indices and paving the way for a new research stage. At that stage, it will be possible to compare and combine short- and ultrashort-term HRV indices with long-term HRV indices to provide a more comprehensive understanding of the autonomic nervous system's response to various stimuli under free-living conditions. The progress in these areas is both promising and exciting.
Point 2:
Are the results of HRV on MDD patients valid compared to other research on MDD? In addition to classification accuracy by machine learning, physiological interpretation is required.
Response:
The HRV indices obtained in our study can be found in Table 4 of the manuscript. Although we had compared these results with previous studies, we acknowledge that this comparison was not sufficiently represented in the manuscript. We appreciate your valuable point regarding the importance of assessing the validity and reliability of HRV indices with prior studies and providing a physiological interpretation. As a result, we have now incorporated a detailed comparison and discussion in the relevant section (page 12, lines 411-430).
Reviewer 2 Report
This is a well written paper on the screening MDD with E4 HRV data. The paper reviewed in depth library of related works and solved some of the key issues in this field.
Though many related works exist, their approach is better and unique in that their work has better result after SQI-FD filtering. Also, thanks to their data collected for the study, this paper can play guiding role for the further study on MDD with PPG sensors.
Comments :
- Definition of Phases of Figure 3 should be included
A few spots in English issues.
- Gold rule => golden rule
- MacBook Air(MI.... ) => MacBook Air
Author Response
We sincerely appreciate your thoughtful and constructive feedback on our manuscript. In response to the comments and advice provided, we have carefully revised the manuscript and offer the following detailed responses to address your queries.
Reviewer #2:
Comment 1: Definition of Phases of Figure 3 should be included
Response:
We greatly appreciate your pointing out this critical issue. We have revised the manuscript in response to your remarks. Specifically, we have added the definition of Phase on page 8 of the manuscript, lines 298-303.
Comment 2: A few spots in English issues.
- Gold rule => golden rule
- MacBook Air(MI.... ) => MacBook Air
Response:
Thank you for taking your valuable time to read this in detail. We apologize for not noticing this before posting. We have corrected the relevant sections (page 2, line 66, and page 9, line 336).
Reviewer 3 Report
The description presented in the article concerns the algorithm for determining characteristic signatures allowing for early detection of MDD syndrome. The analyzed signal is information obtained from a smart watch measuring heart rate variability. Evaluating the article from the point of view of the usefulness of the presented solution, it is very difficult to make an objective assessment. Firstly, the trials were carried out on a very small population of patients, which immediately calls into question the usefulness of the conclusions drawn. Secondly, the subject of the presented article concerns obvious medical problems, where the typical reader of Sensors will not be able to evaluate the proposed solutions. It is true that the applied mathematical tools are adequate to solve such problems, but the lack of a reliable confrontation with the opinions of doctors is a serious lack of the reviewed article.
Author Response
We sincerely appreciate your thoughtful and constructive feedback on our manuscript. In response to the comments and advice provided, we have carefully revised the manuscript and offer the following detailed responses to address your queries.
Reviewer #3:
Point 1:
Evaluating the article from the point of view of the usefulness of the presented solution, it is very difficult to make an objective assessment. Firstly, the trials were carried out on a very small population of patients, which immediately calls into question the usefulness of the conclusions drawn.
Response:
We sincerely thank you have pointed out a crucial issue. There are various limitations to this study. First, there are various individual differences between MDD patients and healthy volunteers. The data of HRV indices will differ from day to day. Therefore we need a considerable size of data to show reliable results. Unfortunately, we could not find any available big data, including continuous 24-hour physiological data from wearable devices and physicians' diagnoses of MDD patients.
In this study, we focused primarily on the issue that the short-term HRV indices measured by wearable devices are sensitive to noise and are reliable as a valuable index for MDD screening only after affected data were removed. SQI-FD proposed in this study is based on the hypothesis that HRV indices calculated from data with the properties that "heart rate cannot be stably estimated" or that "harmonics of the heart rate component buried in noise" are not reliable. Although our study had a limited sample size, we obtained statistically significant results. However, we have not argued that the proposed method would enable classifying MDD patients in any condition. In particular, the influence of antidepressants, described on page 13, lines 461-465 of the manuscript, is the biggest concern and must not be negligible.
The number of subjects in this study was determined a priori according to similar studies [18,19,44]. We carefully examined the obtained data for data leakage and overfitting issues in employing the machine learning method, even for the small sample size. Since it is difficult for our research group to collect a large amount of data shortly, we will consider publishing the proposed method as open source in the future as an alternative measure to make it available to third parties.
Point 2:
Secondly, the subject of the presented article concerns obvious medical problems, where the typical reader of Sensors will not be able to evaluate the proposed solutions. It is true that the applied mathematical tools are adequate to solve such problems, but the lack of a reliable confrontation with the opinions of doctors is a serious lack of the reviewed article.
Response:
We thank and agree with the opinions we have received. As a medical issue, we believe that referring to credible opinions from physicians and developing tools that are useful to them will benefit research and ultimately benefit patients.
We determined our study protocol in consultations with psychiatrists who are co-authors, i.e., Dr. Shinba and Dr. Kariya. They have previously proposed and validated the screening method of MDD patients. The proposed method in this study is an extension of that idea to fit wearable devices. They expect that our system will be a tool to objectively assess the condition of patients at home who are attending intermittently. Because patients who visit clinics continuously and frequently, such as the participants in the return-to-work program included in this study, are less common in real-world clinical practice.
The machine learning model used for MDD screening in this study is a linear model and uses a limited number of features; a discriminant score, a linear combination of HRV indices, is provided, which is not difficult for physicians to interpret. Our co-author Dr. Kariya, who runs a return-to-work program for MDD patients, has often experienced a drop in logit scores when making return-to-work decisions during interviews.
The following research findings are valuable to contrast with physicians' opinions. First, in our previous studies [63, 64], the discrimination scores obtained by our screening system were consistent with physicians' return-to-work decisions in the MDD patients group participating in a return-to-work program. Second, compared to another study[30], discrimination scores correlate more with Zung SDS scores as the screening accuracy improves (lines 380-381 in page 11).
Reviewer 4 Report
The article is well presented and delivers the presentation of the results. There are minor comments from my side, which the authors can address in a review round.
1) Section 2.2: please outline the rationale for the choice of MUSIC frequency estimator, in lieu of other estimators which are simpler (e.g. FFT) pr more resilient (e.g. Buneman).
2) Section 2.2: did the authors define the dB as 20*log10(X)? In general, it is customary to use the definition of 10*log10(X), unless otherwise specified. In this case, please specify which dB definition was used.
3) Only a suggestion: the paper has many acronyms from the very beginning, which is really hard to follow. I suggest the authors include a table of all acronyms either after the intro or as an appendix.
4) Please list authors contribution accordig to the CRediT format
Author Response
We sincerely appreciate your thoughtful and constructive feedback on our manuscript. In response to the comments and advice provided, we have carefully revised the manuscript and offer the following detailed responses to address your queries.
Reviewer #4:
Point 1:
Section 2.2: please outline the rationale for the choice of MUSIC frequency estimator, in lieu of other estimators which are simpler (e.g. FFT) pr more resilient (e.g. Buneman).
Response:
We are delighted to receive your comment based on deep insight into biological signal processing. We have added an supplementary explanation on page 5, lines 245-247 of the manuscript. The calculation of SQI-FD are based on the power spectrum calculated by FFT, so we considered that a more robust method for the noise than FFT was necessary because the calculation result of the signal quality index can be changed by incorrectly selecting the fundamental frequency. And we judged that a further robust approach was unnecessary because data with such low SNR that frequency estimation by the MUSIC method was difficult would be filtered out before the HRV calculation by SQIprv thresholding and the final result would be almost identical.
Point 2:
Section 2.2: did the authors define the dB as 20*log10(X)? In general, it is customary to use the definition of 10*log10(X), unless otherwise specified. In this case, please specify which dB definition was used.
Response:
Thank you very much for pointing out this critical issue. It seems that we have mistakenly described it in terms of the voltage ratio. Since this is a power spectrum, the power ratio is correct, and we have corrected it to 5 dB. Please see the relevant sections: line 214 and the comment in Algorithm 1 on page 5.
Point 3:
Only a suggestion: the paper has many acronyms from the very beginning, which is really hard to follow. I suggest the authors include a table of all acronyms either after the intro or as an appendix.
Response:
We agree that this is also a valid point, and we have included a table for abbreviations in the Appendix. (manuscript pages 14, lines 506-508)
Point 4:
Please list authors contribution accordig to the CRediT format
Response:
The author contribution section has been rewritten in a CRediT-compliant format (page 13, lines 473-480). The many helpful suggestions have greatly improved the quality of the paper. We sincerely appreciate your great support.
Round 2
Reviewer 1 Report
The authors have improved the manuscript by appropriately reflecting my opinion.
Reviewer 3 Report
Thanks for the explanations, I have no further comments